# The Impact of Syntactic and Semantic Proximity on Machine Translation with Back-Translation

**Nicolas Guerin**                                                                 *nicolas.p.guerin@gmail.com*
*Laboratoire de Sciences Cognitives et Psycholinguistique*
*École Normale Supérieure*
*PSL University*

**Emmanuel Chemla**                                                                *emmanuel.chemla@ens.psl.eu*
*Laboratoire de Sciences Cognitives et Psycholinguistique*
*École Normale Supérieure*
*PSL University*

**Shane Steinert-Threlkeld**                                                              *shanest@uw.edu*
*Department of Linguistics*
*University of Washington*

**Reviewed on OpenReview:** *https://openreview.net/forum?id=6DflIABPQP*

## Abstract

Unsupervised on-the-fly back-translation, in conjunction with multilingual pretraining, is the dominant method for unsupervised neural machine translation. Theoretically, however, the method should not work in general. We therefore conduct controlled experiments with artificial languages to determine what properties of languages make back-translation an effective training method, covering lexical, syntactic, and semantic properties. We find, contrary to popular belief, that (i) parallel word frequency distributions, (ii) partially shared vocabulary, and (iii) similar syntactic structure across languages are not sufficient to explain the success of back-translation. We show however that even crude semantic signal (similar lexical fields across languages) does improve alignment of two languages through back-translation. We conjecture that rich semantic dependencies, parallel across languages, are at the root of the success of unsupervised methods based on back-translation. Overall, the success of unsupervised machine translation was far from being analytically guaranteed. Instead, it is another proof that languages of the world share deep similarities, and we hope to show how to identify which of these similarities can serve the development of unsupervised, cross-linguistic tools.

## 1 Machine translation and the role of back-translation to eliminate supervision

Supervised training for neural machine translation (NMT) requires a vast amount of parallel data (Sutskever et al., 2014; Wu et al., 2016; Vaswani et al., 2017). Creating the necessary datasets aligned across languages is a complex, onerous and sometimes impossible task. In this context, unsupervised back-translation, and in particular on-the-fly back-translation, becomes highly valuable, as it allows unsupervised training from independent monolingual corpora (Sennrich et al., 2016; Lample et al., 2018a; Guzmán et al., 2019; Haddow et al., 2022).

Back-translation works by using a translation model from one language ($L_1$) to another ($L_2$) to synthetically generate data in the following way: a given text $x$ in $L_1$ is passed in and a hypothetical translation $\tilde{y}$ in $L_2$ is generated. This pair is then treated as if it were a 'gold' translation $(\tilde{y}, x)$ to train an $L_2$-to-$L_1$ system. For iterative and on-the-fly back-translation, the whole process is repeated in the other direction, and iterated, so that the models improve as they are generating the data. One can think of this process, in essence, as

that of an auto-encoder of one of the languages, with the hidden space being the other language: the model is trained to translate a sentence from language 1 into language 2 and back into language 1, success being attained if the original and final sentence from language 1 match.

Back-translation was originally used as data augmentation, i.e. part of the training was also done on a parallel corpus (Sennrich et al., 2016). In the more recent literature some models are fully trained via back-translation in an unsupervised way on top of other unsupervised objectives such as denoising auto-encoding or language modeling, and obtain decent BLEU scores, with significant improvements for low-resource languages (Lample & Conneau, 2019; Liu et al., 2020, i.a.). Here, then, success is obtained without any training signal as to how the two languages are aligned. This empirical success, however, is puzzling: without explicit information about how to align the two languages, how does the method succeed? To our knowledge, there is no systematic understanding as to why this type of training succeeds.

In this paper, we first discuss why the success of back-translation is puzzling in general (Section 3). We conjecture, as many do, that it works because, despite surface differences, natural languages have rich and similar structures that can promote alignment (Lample & Conneau, 2019; Wu & Dredze, 2019; Conneau et al., 2020b). We then describe our shared exerpimental setup (Section 4), which uses artificial languages to systematically manipulate various similarities and differences between languages. We then report a series of experiments which analyze which properties drive the observed empirical success of back-translation (Sections 5-10). Our systematic experiments suggest that shared syntactic and very simple semantic structure does not suffice for back-translation to yield quality unsupervised NMT (UNMT) systems. Hence, shared *complex* semantic dependencies appears to be crucial.

Our contributions are: (i) an analysis of why back-translation should not work in general, (ii) the use of artificial languages to conduct systematic experiments on factors driving its empirical success, and (iii) the elimination of reasons (often claimed to be behind its success) such as close syntactic structure, word frequency, anchor-points or even a semantic structure of lexical fields.

## 2 Related work

Prior to unsupervised translation, several works focused on bilingual dictionary induction but using little or no parallel data. Mikolov et al. (2013) does so by relying on a distributed word representation of large monolingual corpora as well as a small supervised corpus. Klementiev et al. (2012); Irvine & Callison-Burch (2016) develop a phrase-based statistical model for phrasal and lexical translation. More recently, Lample et al. (2018b) allows dictionary inference without any parallel data.

Bojar & Tamchyna (2011) used *reverse self-training* to train a model in a statistical machine translation (SMT) framework. The idea is to use a small parallel corpus to train a machine translation (MT) system from target to source and use it to translate a large monolingual target-side corpus to create synthetic pairs to train a source to target SMT model. This is the root of Back-translation (BT) as a data augmentation method. This is applied to NMT by Sennrich et al. (2016) who train two models iteratively, source to target and target to source. The idea that translation from a language to another has a dual task, which is the translation in the reverse order, that can be learned jointly to improve efficiency is largely used in several works (Xia et al., 2017; Cheng et al., 2016; He et al., 2016; Wang et al., 2019; Hoang et al., 2018; Niu et al., 2018). For example, Cotterell & Kreutzer (2018) use the formalism of a wake-sleep algorithm to propose an Iterative Back-Translation method, i.e. the synthetic data is regenerated after each model training. However, in all these works training is still partially supervised and back-translation acts as data augmentation.

Artetxe et al. (2018) introduce a fully Unsupervised Neural Machine Translation (UNMT) method. It uses on-the-fly BT (OTF-BT), meaning that synthetic sentences are generated as the model trains. They also add a denoising autoencoding objective. This paper lays the foundations for the following BT-based architectures and models. For example, Lample et al. (2018a) use the same method and leverage the intuition that the same sentence, regardless of the language, will be mapped onto the same embedding vector by encoders, and that to translate the sentence it is then sufficient to decode it into the desired language. They enforce this behavior using an additional adversarial training objective. Other works remove this adversarial objective

while maintaining performance (Lample et al., 2018c; Yang et al., 2018), suggesting that this embedding space overlap emerges naturally.

Following on from this line of work, several papers formulate the idea that the quality of the cross-lingual embeddings used to initialize BT are key and therefore add a language modeling objective as pre-training (Lample et al., 2018a; Edunov et al., 2019). This method works very well and has become the standard approach in UNMT. For example, Liu et al. (2020) introduces mBART for UNMT while Zan et al. (2022) studies in depth its benefits. Numerous papers propose modifications to BT to improve its performance, particularly in a low-resource setup (Kumari et al., 2021; Kim et al., 2019; Pourdamghani et al., 2019; Dou et al., 2020; Song et al., 2019). In particular, Filtered BT (Khatri & Bhattacharyya, 2020; Imankulova et al., 2017) filters synthetic sentences based on their quality, using a round-trip sentence similarity metrics, while Tagged BT (Caswell et al., 2019) prepends a tag in front of synthetic sentences to indicate to the model which are from the parallel corpus and which are not.

Many studies examine the reasons for the success of BT for UNMT (Edunov et al., 2018; Poncelas et al., 2018; Kim et al., 2019; Edunov et al., 2020; Conneau et al., 2020b). In particular, some focus on the properties of languages, pursuing the thought that BT and multilingual models work due to the similar structures of natural languages. K et al. (2019) studies the impact of several factors on the cross-lingual abilities of mBERT, including the linguistic properties of languages, such as token overlap between vocabularies, or word order. They show that the former has very little effect, but that the latter is fundamental. Dufter & Schütze (2020) takes up the same kind of work. Finally, Kim et al. (2020) shows that UNMT performance drops drastically when languages or domains are very different. It also shows that BT cannot recover from a bad initialization.

Our work continues this line of research, but differs in that we use artificial languages to maintain perfect control over our experiments and to avoid confounding factors as much as possible. This allows to investigate precise syntactic and semantics effects on OTF-BT. We also use these artificial languages to demonstrate the inaccuracy of the usual assumptions concerning the success of OTF-BT (e.g. word frequencies, anchor points, syntax, etc.). To the best of our knowledge, this has not been done in previous works. It is hoped that this work will lead to greater intuition about BT, enabling researchers to refine the method and produce better results in the future.

Several other works also use artificial languages on related tasks (Arora et al., 2016; White & Cotterell, 2021; Chiang & yi Lee, 2021), but none apply their methodology to UNMT. Finally, it should be noted that the model used and studied in this work (Lample et al., 2018c) is no longer state of the art, but uses OTF-BT almost exclusively, unlike more recent models which rely heavily on massive pre-training, making it extremely relevant to our work.

## 3 Back-Translation is necessary, but not sufficient in general

A fully unsupervised training with back-translation works as follows. The model consists of classical components of a translation system, as seen in Figure 1. It includes encoders $e_i$ from language $L_i$ into a hidden space, and decoders $d_i$ from this hidden space into $L_i$. These encoders and decoders could be separate models or the same multilingual model conditioned on a language ID token (Conneau et al., 2020a; Liu et al., 2020). The composition of an encoder $e_i$ and decoder $d_j$ from different languages provides a translation function $T_{ij}$.

These components are first trained through a denoising auto-encoding, monolingual regime, whereby $d_i \circ e_i \circ g$, with $g$ a noise function, should lead to the identity for each language. Then suppose $y$ is an element of the language $L_2$. We can translate $y$ into $L_1$ as $\tilde{x} = d_1 \circ e_2(y) := T_{21}(y)$. This creates a synthetic pair $(\tilde{x}, y)$ in $L_1 \times L_2$. This pair, which was created by the model itself, is now used as a supervised datum for translation. That is, we check that the translation (now backwards) from $\tilde{x}$ in $L_1$ to $L_2$ is coherent: $\tilde{y} = d_2 \circ e_1(\tilde{x}) := T_{12}(\tilde{x})$

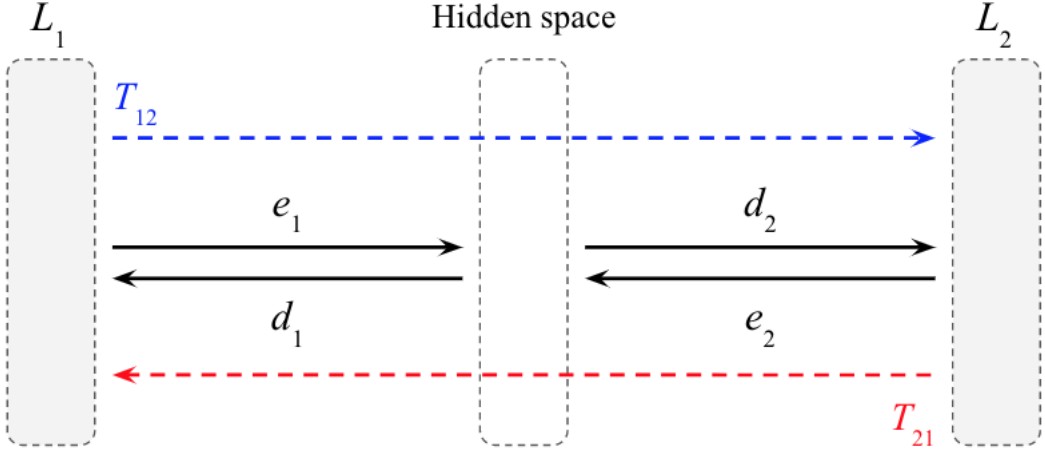

Figure 1: The components of a classical translation pipeline. Translation functions $T_{12}$ and $T_{21}$ are obtained from the composition of encoders and decoders from different languages.

should lead back to $y$. With a loss function $\mathcal{L}$ (e.g., cross entropy), this process can be summarized as:

$$
\begin{aligned}
\min_{\theta_{e_i,d_i}} \sum_{y \in L_2} &\mathcal{L}(y, d_2 \circ e_1 \circ d_1 \circ e_2(y)) \\
&+ \sum_{x \in L_1} \mathcal{L}(x, d_1 \circ e_2 \circ d_2 \circ e_1(x)) \\
&+ \sum_{j \in \{1,2\}} \sum_{z \in L_j} \mathcal{L}(z, d_j \circ e_j \circ g(z))
\end{aligned}
\tag{1}
$$

where $\theta_{e_i,d_i}$ are the parameters of the encoders/decoders. The gray components in the first two lines reflect 'frozen' parts of the pipeline: in practice we sample from those $d_i$ and then pass those generations forward, as previously described. Hence the formula is a slight simplification. In practice, the denoising autoencoding objective (or similar; see third line in (1)) often happens *before* the back-translation objective is used; we have included them as one loss because the present arguments do not depend on this choice.

With this framework in place, we can now explain why back-translation is a necessary objective, but not a sufficient one. The back-translation objective (1) could be fully met, without translation being accurate at all. We illustrate this visually in Figure 2, with two extreme cases.

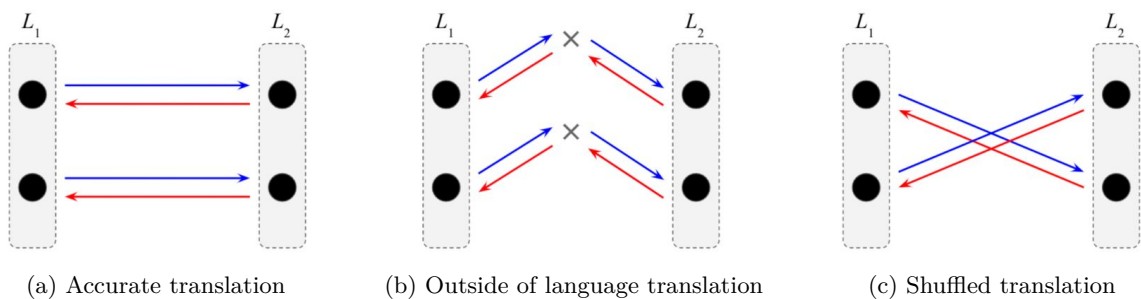

(a) Accurate translation  (b) Outside of language translation  (c) Shuffled translation

Figure 2: Schematic representation of three cases in which the back-translation objective (1) is fully met: (a) with an accurate translation, (b) with a translation missing the target language entirely, but being sent back on the original language appropriately, (c) with a translation that bijectively shuffles the target language, and a reverse translation that unshuffles it back in place. We ignore here the hidden encoding space, and write $\longrightarrow$ for $T_{12}$ and $\longleftarrow$ for $T_{21}$.

First, consider Figure 2b. It could be that the "translation" $T_{12}$ from $L_1$ to $L_2$ projects $L_1$ into a space $X$ outside of the actual $L_2$ (so translation will be completely off), but that the back-translation would project this space $X$ back into $L_1$ appropriately.

Second, consider Figure 2c. It could be that the space $X$ overlaps well with $L_2$, but that it is a shuffled version of it, in such a way that translation will be completely off again. Yet, the back-translation objective will be fully met, as long as the translation un-shuffles it back in place onto $L_1$.

More generally, any combination of $e_i$ and $d_i$ such that $T_{12} = T_{21}^{-1}$ verifies the back-translation component of (1); this essentially requires that the translation functions $T_{ij}$ be inverse bijections, but such bijections do not need to correspond to accurate translation functions. The auto-encoding component of (1) does not suffice *a priori* to overcome this problem.

These simple counter-examples show how back-translation could fail. In practice, however, back-translation has achieved significant empirical success (Lample et al., 2018a; Lample & Conneau, 2019; Liu et al., 2020; Song et al., 2019, i.a.). Back-translation must work because natural languages look like one another, i.e. structure in the data makes it so that the proper alignments between $L_1$ and $L_2$ are easier to discover than the external ones or shuffled ones. But what structure in the data? The intuition often put forward is that back-translation works because, e.g., frequent words are better mapped onto frequent words, or because systematic word orders will anchor the mappings. In the rest of this paper we investigate such hypotheses in more detail.

## 4 Experimental Setup

We investigate which language properties make back-translation work or fail. To do this, we tested back-translation between artificial languages for which we can freely manipulate lexical, syntactic and semantic properties.

### 4.1 The Context Free Languages

The grammars of our artificial languages vary somewhat similarly to what is assumed for natural languages. We used the simple Context Free Grammars (CFG) introduced by White & Cotterell (2021), which are parametrized by 'switches': the order in which constituents surface. Table 1 presents the rules that are switchable in these grammars. We will denote a grammar by a sequence of six binary values, corresponding to the switches. These syntax rules are designed to reflect real rules that differentiate many natural languages. For example, Japanese corresponds to the combination 000000, English to 011101 and Spanish to 011111.

| | Rules for each switch value | |
|---|---|---|
| **Switch** | **0** | **1** |
| **S** | S $\longrightarrow$ NP VP | S $\longrightarrow$ VP NP |
| **VP** | VP $\longrightarrow$ NP VP | VP $\longrightarrow$ VP NP |
| **Comp** | $S_{Comp} \longrightarrow$ S Comp | $S_{\mathbf{Comp}} \longrightarrow$ Comp S |
| **PP** | NP $\longrightarrow$ PP NP | NP $\longrightarrow$ NP PP |
| | PP $\longrightarrow$ NP Prep | PP $\longrightarrow$ Prep NP |
| **NP** | NP $\longrightarrow$ Adj NP | NP $\longrightarrow$ NP Adj |
| **Rel** | NP $\longrightarrow$ VP Rel Noun | NP $\longrightarrow$ Noun Rel VP |

Table 1: Rules that are switchable in the grammar. Table from White & Cotterell (2021).

At the lexical level, 1,374 words were created, with a plausible English morphology, and distributed in several Parts-of-Speech (POS), based on the work of Kharkwal (2014). We used 1,374 words to obtain a replication of White & Cotterell (2021)[1]. Picking a set of switches (i.e. a grammar) and a set of such created words thus

---

[1]Many of our points rely on the *failure* of OTF-BT (see experiments), hence we did not try with a larger vocabulary, which would be a harder task and could only confirm failure.

defines one artificial language. A full translation between two such simple languages can be reconstructed with a one-to-one mapping at the lexical level, and knowledge of the relevant switches.

We provide some examples of sentences generated by such artificial languages in Table 2, illustrating the impact of the vocabulary changes and the grammatical switches.

| Language | Examples |
|---|---|
| **000000 POS** | NounS Subj VerbCompPresS. |
| **000000 (Lexicon 0)** | burse sub lurchifies. |
| **100000 (Lexicon 0)** | lurchifies burse sub. |
| **100010 (Lexicon 0)** | lurchifies burse sub. |
| **000000 (Lexicon 1)** | swopceer bus rheleates. |
| **000000 POS** | IVerbPastP Adj NounP Subj. |
| **000000 (Lexicon 0)** | rolveda prask autoners sub. |
| **100000 (Lexicon 0)** | prask autoners sub rolveda. |
| **100010 (Lexicon 0)** | autoners sub prask rolveda. |
| **000000 (Lexicon 1)** | knyfeateda wourk krarfteers bus. |

Table 2: Examples of sentences generated by our artificial languages. The POS row gives the underlying structure of the sentence in each group (colors in the following rows trace these Part-of-Speech). When relevant, constituents swapped compared to the grammar from the row above are underlined. The grammars and switches are described in Table 1.

## 4.2 Training

**Model**  We used the exact model introduced by Lample et al. (2018c) for NMT. The architecture is based on 4 Transformer encoder layers and 4 Transformer decoder layers. The last 3 encoder layers and the first 3 decoder layers are shared across languages. The embedding size is 512 and the vocabulary, of size $1,000$, was shared and built using Byte Pair Encoding. We use this model because of its high-performance (state of the art at the time of publication) and because it allows us to isolate the study of OTF-BT.

**Data**  For training purposes, we generated two sets of 100,000 sentence structures[2]. All unsupervised training sets are made of (i) one of these sets of sentence structures for the 100,000 sentences in one language (using the language specific grammar and lexicon), and (ii) the other set to create 100,000 sentences in the other language. Hence, the data are neither labeled for supervision, nor are they even unlabelled translations from one another in principle. The test and validation sets are each composed of 10,000 parallel sentence pairs. These are generated in one language, and then transformed into the second language using the known grammar switches and lexical translations.

**Procedure**  At the start of training, a FastText algorithm is applied to both languages simultaneously to initialize the cross-lingual embeddings (Bojanowski et al., 2017). We then train the translation model for 40 epochs, with a batch size of 16 and an Adam optimizer with a $10^{-4}$ learning rate. Those hyper-parameters were chosen based on Lample et al. (2018c) and on our computational capabilities. We trained on a Tesla M10 with 8GB memory for roughly one day.

**Objectives**  The overall training objective is described in (1). First the model does a denoising auto-encoding (DAE) step, which can be seen as a LM step that constrains the model to output in the correct language, and updates its parameters, then it does an on-the-fly back-translation step in both directions and updates its parameters again. The denoising function $g$ used is a combination of word substitution, local shuffling and masking. We note that DAE is here interspersed with BT, not used as a separate pretraining objective.

---

[2]Using up to a million sentences yielded no improvement in all instances that we tried, so we report results for 100,000 sentences to save resources.

**Evaluation** In each of the next experiments the results obtained are those computed on the test set by the model having obtained the best BLEU score on the validation set. Note that the BLEU score calculated on our artificial languages is not to be compared with BLEU scores obtained on natural languages with much larger vocabularies and many turns of phrase that count as correct translations. However, as a point of reference, the Lample et al. (2018c) model we use obtained on WMT monolingual News Crawl a BLEU score of 24.65 on English-French pairs, and of 19.10 on English-German pairs which was UNMT state of the art.

### 4.3 Experiments

In the remainder of this paper, we present six experiments, each of which will answer several research questions, some of which will emerge from the results.

The first experiment §5 is very simple and acts as a sanity check, the idea being to ensure that the model is indeed capable of learning to translate these artificial languages into themselves.

The second experiment §6 is to measure the impact of grammar on translation performance, keeping the lexicon equal. In this way, the translation function is only concerned with the change in grammar from one language to another.

The third experiment §7 adds the difficulty of a new lexicon. We show that this difficulty completely undermines the success of the machine translation system. This, of course, is different from the situation with real-world languages.

The following three experiments then seek to understand what additional signal is present in natural languages and is so crucial to the success of OTF-BT. Experiment four §8 adds anchor words and identical word frequency. Experiment five §9 shows that a weak supervised signal, such as a bilingual dictionary or a small supervised dataset, restores the model's performance. Finally, experiment six §10 adds semantic information by way of lexical fields.

### 4.4 Metrics

We use the BLEU score as a metric of accurate translation (Papineni et al., 2002, and in particular the Moses implementation as in Lample et al., 2018c).[3] Some more specific metrics will be introduced for the particular of some experiments, but BLEU will serve as a standard reference in many places. There are various limitations of BLEU: it cannot handle paraphrases and synonyms, does not take meaning into account directly, it is invariant by n-grams permutations, it has a low correlation with human judgment, etc. The limitations concerning paraphrases and synonyms do not apply to our setup, which does not have paraphrases nor synonyms. Similarly, meaning is not relevant in our setup. Overall, BLEU ignores certain aspects and may lead to overestimations of the quality of translation, To mitigate this, we computed BLEU scores for n-grams ranging from 1 to 4. This helps capture syntactic structure (order constraints) and this mitigates the risk of bigrams permutations insensitivity, as illustrated in Callison-Burch et al. (2006). In any event, however, our arguments rely mainly on the observation of *low* BLEU scores, hence overestimation is a mild risk.

Alternatives to BLEU include two families: the automatic language agnostic metrics like BLEU (Papineni et al., 2002), METEOR (Banerjee & Lavie, 2005) or Word Error Rate (WER), and the LLM-based metrics like BERTScore (Zhang et al., 2019), MetricX (Juraska et al., 2023), Comet Rei et al. (2020), Unite (Wan et al., 2022), BLEURT (Sellam et al., 2020), etc. The second family of measures is not applicable to our artificial languages. Among the first family, WER is known to be worse than BLEU, METEOR remains imperfect (see Saadany & Orasan, 2021) and BLEU therefore seems like the right choice.

---

[3]https://github.com/moses-smt/mosesdecoder/blob/master/scripts/generic/multi-bleu.perl

## 5  Experiment 1: Identical $L_1$ and $L_2$ languages

Our first experiment demonstrates that our hyper-parameters and training pipeline work in principle, by using BT to translate between two identical languages. Note however that, in principle, even for identical languages, back-translation is not a sufficient objective. We randomly selected 8 grammars among the 64 possible ones, and picked one lexicon, to form 8 artificial languages. For each such language $L_1$, we trained our model to learn translation between $L_1$ and a similarly defined language $L_2$. Sampling of training sets was done independently for the two identical languages, hence the two monolingual corpora are not aligned.

Table 3 reports BLEU scores on the test set in this paradigm. Training was successful: all BLEU scores are above 97.

| Grammars | BLEU |
|---|---|
| 000000↔000000 | 98.76 |
| 011101↔011101 | 98.70 |
| 011111↔011111 | 99.08 |
| 000001↔000001 | 98.13 |
| 100000↔100000 | 98.83 |
| 000101↔000101 | 97.76 |
| 111111↔111111 | 97.77 |
| 111110↔111110 | 97.30 |

Table 3: BLEU scores obtained on the test set for within grammar and within vocabulary training. Those scores are the average between the BLEU score obtained when evaluating source to target and target to source respectively.

White & Cotterell (2021) found that the choice of the grammar had an impact on language modeling success for the Transformer architecture (but not for LSTM language models). Here as well, we found that the choice of grammar had an effect on the result, and in the same way: our within-language BLEU scores varied and were correlated with the perplexity on the same languages obtained in their language modeling task ($R^2 = 0.62$). In short, some word orders robustly lead to better performance both for language modeling and, now, for within-language translation.

## 6  Experiment 2: Effect of Grammar

If similar structure is the key for back-translation to work, then translation between two languages should be harder if their grammar are more different (even if we stay within the variation observed between actual languages). We can operationalize this hypothesis in terms of the Hamming distance (number of different switches) between the grammars generating two languages: we expect BLEU score to decrease as Hamming distance increases. In these tests, we keep the vocabulary constant across languages to focus on the effect of grammar.

To test this hypothesis, we used the eight random grammars of §5 and trained a translation system via back-translation for each pair, resulting in 64 BLEU scores. The complete results are in Table 4.

First, we found a correlation between the obtained BLEU scores and the Hamming distance ($R^2 = 0.35$ with a coefficient of $-8.35$, $p < 0.001$). In other words, back-translation does become less performant as the grammars differ more.

Second, we used as predictors not the raw Hamming distance, but individual variables $S_i$ corresponding to each switch being different or not in the translation pair at stake, thus fitting a model of the form **BLEU** $\sim \sum_{i=1}^{6} \beta_i S_i$. We obtain a strong fit ($R^2 = 0.94$), and a significant effect at $p < 0.001$ for the coefficients corresponding to switch $S_1$ ($\beta_1 = -45.04$), $S_4$ ($\beta_4 = -6.36$) and $S_6$ ($\beta_6 = -11.04$). This suggests that not all switches are created equal: some have more dramatic effects than others on back-translation performance.

| | 000000 | 011101 | 011111 | 000001 | 100000 | 000101 | 111111 | 111110 |
|---|---|---|---|---|---|---|---|---|
| **000000** | 98.8 | 64.3 | 67.8 | 73.5 | 46.8 | 68.3 | 35.5 | 36.5 |
| **011101** | 64.7 | 98.7 | 98.6 | 87.4 | 36.8 | 96.5 | 38.7 | 37.9 |
| **011111** | 67.8 | 98.7 | 99.1 | 86.2 | 36.1 | 93.9 | 39.4 | 40.4 |
| **000001** | 73.9 | 85.6 | 83.3 | 98.1 | 39.9 | 91.8 | 29.3 | 34.5 |
| **100000** | 43.9 | 33.8 | 33.7 | 33.4 | 98.8 | 31.3 | 63.1 | 76.0 |
| **000101** | 69.4 | 90.3 | 96.7 | 91.8 | 37.6 | 97.8 | 34.1 | 40.5 |
| **111111** | 41.6 | 40.8 | 41.9 | 36.1 | 67.1 | 40.5 | 97.8 | 86.6 |
| **111110** | 40.6 | 36.4 | 38.5 | 39.2 | 78.3 | 46.1 | 86.0 | 97.3 |

Table 4: BLEU scores for languages with different grammars but the same lexicon.

Because of this, we more systematically evaluated the effect of each individual switch. To do this, we use the 000000 grammar as the source language and vary the target language so that one switch only is activated each time: 100000, 010000, ..., 000001. Conversely, we used 111111 as the source language, deactivating each switch in turn for the target language: 011111, 101111, ..., 111110. The results are in Table 5.

| Grammars | BLEU | id baseline | | Grammars | BLEU | id baseline | |
|---|---|---|---|---|---|---|---|
| **000000↔1**00000 | 45.36 | 51.04 | (-0.11) | **111111↔0**11111 | 41.25 | 48.23 | (-0.17) |
| **000000↔0**1**0000** | 94.81 | 42.37 | (1.24) | **111111↔1**0**1111** | 87.94 | 43.27 | (1.03) |
| **000000↔00**1**000** | 95.93 | 70.33 | (0.36) | **111111↔11**0**111** | 97.86 | 69.57 | (0.41) |
| **000000↔000**1**00** | 92.88 | 94.63 | (-0.02) | **111111↔111**0**11** | 92.59 | 93.95 | (-0.01) |
| **000000↔0000**1**0** | 98.69 | 82.60 | (0.20) | **111111↔1111**0**1** | 97.56 | 79.46 | (0.23) |
| **000000↔00000**1 | 73.17 | 73.17 | (0.00) | **111111↔11111**0 | 86.40 | 69.66 | (0.24) |

Table 5: The BLEU scores show that different switches have different impacts on the translation performance, with source language **000000** (left) and **111111** (right). The second column shows the baseline BLEU score that would be obtained by a translation system that would just copy the initial sentence and, in parentheses, the relative distance of the learned translation to this baseline: $\frac{\text{BLEU}-\text{baseline}}{\text{baseline}}$.

Different switches show different impact on the translation performance, in a way mirroring the regression coefficients: switch 1 causes the largest drop in performance, followed by switch 6 and then switch 4. Other switches, like the fifth one, have little impact on BLEU.

An intuition behind the different impacts of the switches could be as follows: the first switch governs a change that occurs near the root of the parse tree while the fifth switch concerns a node closer to the leaves. Thus, more words move, and they move further away in the first case than in the second. Because of this, a model that has learned the identity mapping (and therefore has not learned the translation) would get a higher BLEU score with the fifth switch than with the first.

To isolate this effect and determine whether the drops in BLEU are a result of a bias towards learning an identity translation, we computed the BLEU score of a system that would just copy the source sentence as is (and the relative distance between the BLEU score obtained by our model and this baseline). The results are shown on the last two columns of Table 5.

Different (relative) effects of each switch on the translation are found again, suggesting a more subtle effect than the intuition mentioned above. In sum, grammar changes that make words move further away have more impact on the translation performance, and this is not because the identity is learned instead of a proper translation.

# 7   Experiment 3: Effect of the Lexicon

Until now, the two languages had the same lexicon but different grammars. In this section we focus on the opposite scenario, i.e. translation between languages with an identical grammar, but different lexica. We thus built a second, parallel lexicon of 1,374 new fictitious words with an English-like morphology using the same methods as before (see §4.1).

We followed the same training procedure on five out of the eight grammars from §5. In this context, we found that back-translation systematically failed to produce successful translation systems: the best BLEU score was below 7, as reported in Table 6. Note that the round-trip BLEU score was above 70 in all trainings. This shows that it's not the back-translation objective that has failed to be optimized, but rather that it's not sufficient to ensure correct translation.

| Grammars | BLEU | POS BLEU |
|---|---|---|
| 000000 ↔ 000000 | 2.46 | 69.3 |
| 011101 ↔ 011101 | 2.81 | 62.4 |
| 000001 ↔ 000001 | 4.31 | 67.3 |
| 100000 ↔ 100000 | 5.68 | 76.2 |
| 111111 ↔ 111111 | 6.63 | 58.1 |

Table 6: BLEU scores obtained on the test set for within grammars training but with different lexica. It is clear that the training did not work as the highest BLEU score is below 7, demonstrating on the fly that back-translation is not guaranteed to succeed. The **POS BLEU** colomn report the average BLEU score obtained on the Part-of-Speech only. These high scores show that the failure lies in the vocabularies mapping.

To identify the source of these errors, we computed BLEU scores for part-of-speech (POS) values instead of exact tokens. In this setting, all BLEU scores were above 60 (see Table 6). In more detail, around a third of the sentences in the test set are syntactically correctly translated. The average length of correct sentences is 6.2, compared with 11.0 for the whole test set. Correct sentences are therefore shorter on average, which is not surprising simply if shorter length means fewer opportunities of error. This relatively good syntactic performance can be partly explained by the DAE objective, which constrains the decoder to output sentences from the grammar. However, it is also clear that the model does not ignore input when doing translation.

To further document the failure at the vocabulary level, we focussed on an analysis of sentences whose POS sequence has been correctly translated; there it is possible to do word-by-word evaluation. For any word $w_s$ from the source language, we looked at the distribution of its translations by the model across its different occurrences in this restricted test set. The perfect model would always translate $w_s$ by its correct translation, and would have an entropy of 0. By comparison, a model choosing the translation at random (although in the correct POS) would have an entropy of $\log_2$ of the size of the relevant POS set. Taking singular names as an example, the average entropy of the model is 0.05, whereas a random model would have an entropy of 7.3. The results for the other POS are qualitatively identical (see Table 7), showing that the model did pick a single translation per word, albeit not the correct one. This points towards the shuffled translation scenario illustrated in Figure 2c for the vocabulary.

# 8   Experiment 4: Unsupervised Vocabulary Signals

Back-translation alone was unable to learn the mapping between fully distinct vocabularies. Here we model two phenomena that could help overcome this difficulty in natural settings: the presence of shared lexical items across languages, or "anchor points" (Conneau et al., 2020b), and similar word frequencies across languages (Piantadosi (2014), among many others).

| POS | Entropy | (random) |
|---|---|---|
| **Noun** | 0.05 | (7.3) |
| **Adjective** | 0.05 | (5.4) |
| **Transitive Verb** | 0.06 | (6.4) |
| **Intransitive Verb** | 0.05 | (6.8) |
| **Verb Complementizer** | 0.03 | (4.5) |

Table 7: Entropy for each POS (in parentheses, entropy value for a random translation within the correct POS). For each POS, the results have been averaged over the different morphological variants. For example, the **Noun** row groups together Singular and Plural Nouns.

| | Manipulation | Exp. | BLEU score |
|---|---|---|---|
| | **Lexical change** | 3, §7 | 2.5 |
| (i) | **Anchor Points** | 4a, §8.1 | 23.0 (+20.6) |
| (ii) | **Frequency** | 4b, §8.2 | 10.1 (+ 7.6) |
| (iii) | $\mathbf{N_c = 2}$ | 6a, §10.1 | 9.2 (+ 6.8) |
| (iv) | $\mathbf{N_c = 2}$ | 6b, §10.2 | 10.7 (+ 8.3) |
| (v) | $\mathbf{N_c = 10}$ | 6c, §10.3 | 11.9 (+ 9.5) |

Table 8: BLEU scores obtained from back-translation, augmented with other unsupervised signals. We start from the previous experiment with a lexical change, then show results when we supplement the situation with (i) anchor points, (ii) (intra-POS) word frequencies (for the **000000** grammar and two different lexica), (iii) $N_c = 2$ equally balanced disjoint lexical fields, (iv) $N_c = 2$ unbalanced disjoint lexical fields, (v) $N_c = 10$ disjoint lexical fields. In parentheses: the gain of BLEU score compared to Exp 3.

### 8.1 Experiment 4a: Anchor Points

In natural language corpora, a number of words are identical between languages, in particular in written form. This happens because the languages have a common root, because some words have been borrowed across languages, because of identical proper nouns, or because the languages may use similar numeral systems. These identical words between two languages may serve as anchor points that allow translation models to fit the whole mapping between two lexica. This could help prevent the rotation problem described in §3 and the disjoint lexicon problem of §7.

To test this hypothesis, we ran back-translation on two languages with the **000000** grammar and lexicons sharing 30% of their words. We obtained a BLEU score of 23.04. Using the word-by-word comparison method from §7 (for syntactically correct translations), we show that 92% of the words with non-zero precision and recall are within the common words. This suggests, mirroring similar findings on multilingual language modeling due to Conneau et al. (2020b), that common words do not serve as good anchors that can lead the empirical success of back-translation on the *entire* vocabulary.

### 8.2 Experiment 4b: Frequency alignment

In the experiments described thus far, all words within a given POS have equal frequency. In natural languages, however, words vary in frequency, and presumably in similar ways across languages, following a power law. This information could help a translation model learn the mapping between the two vocabularies.

We hence manipulated our corpora so that the probability of appearance of words within each POS follows a power law: $P(w_n) \propto n^{-k}$ for $k = 1.1$, as they roughly do in natural languages, where $w_n$ is the $n$-th most frequent word.

With the **000000** grammar and two different lexica following these parallel Zipfian distributions, we obtained a BLEU score of 10.08, a slight increase compared to §7. Using again the word-by-word translation analysis,

we observed no regularity in which words were appropriately translated: it is not the case that the most/least frequent words are better translated, it is not the case that the model outputs the most frequent words. This is illustrated in Figure 3 with singular nouns: most words have a very low accuracy and recall, and those that do not are spread across the x-dimension (rank).

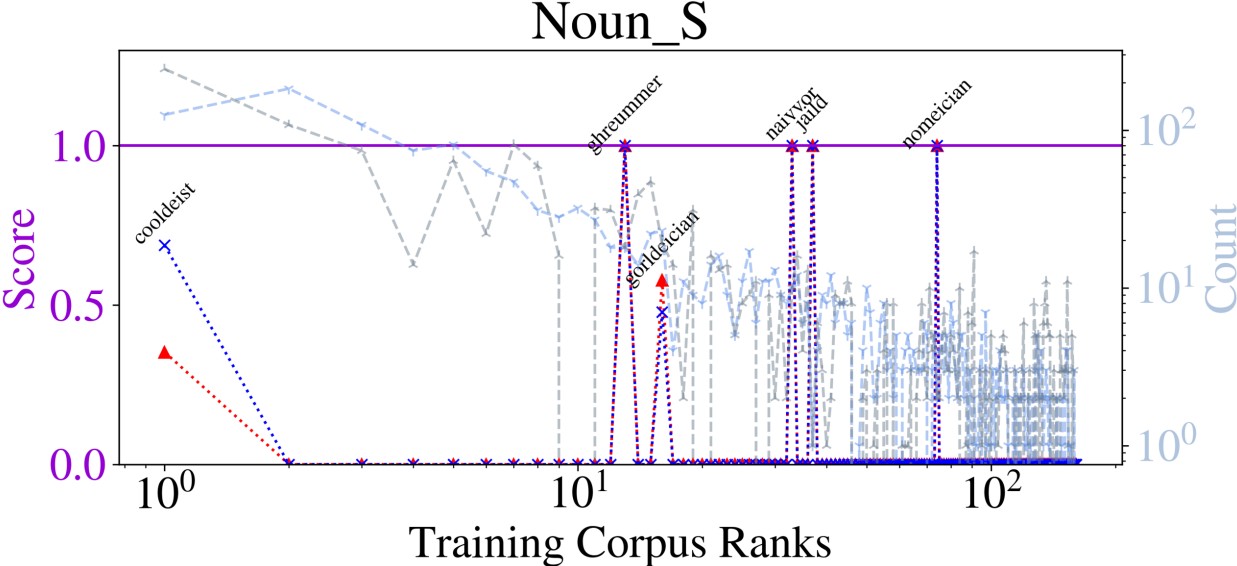

Figure 3: The red (resp. blue) line represents the accuracy (resp. recall) of the lexicon word translation. These are ranked by their frequency in the training corpus. The dashed lines represent word counts in the testset (light blue) and in the model translation (grey). It can be seen that the model tracks actual frequencies well. Then, very few words have a non-zero score, which is consistent with a very low BLEU score. What's more surprising is that the most frequent words are not translated any better.

Thus, in the same way that anchor points did not help much (§8.1), rich and realistic word frequencies cannot be the primary driver of the empirical success of back-translation. This is disappointing, as a simple mapping of lexica based on word frequency in the corpus would yield a very good translation.

Finally, the POS BLEU score for this experiment is 57, with only 22% sentences being correctly translated syntactically (compared to 33% before). In the case where intra-POS frequencies were uniform, word frequencies were therefore fixed by the POS frequency, itself fixed by the grammar. This clustering of word frequencies based solely on syntax may explain the better syntactic score of previous experiments. In other words, rich word frequencies do not improve vocabulary translation, and it impacts translations at the level of syntax.

## 9 Experiment 5: Adding Supervision

One possible explanation for the failures above could be the rotation effect described in §3. Some supervised training could help avoid that risk, by anchoring at least some translations into ground-truth. We test two types of supervised examples: some aligned sentence translations (as when back-translation is used not for fully unsupervised translation but for data augmentation), and the injection of a bilingual dictionary.

### 9.1 Small Aligned Dataset

Here, we use back-translation as data augmentation, as it was originally used, by training the model in a supervised way on 1,000 sentences, on top of the training with back-translation on the same 100,000 sentences from before. The training process is now as follows: first a step of denoising auto-encoding, then a

supervised step with a batch from the aligned dataset and finally a back-translation step with a batch from the unaligned dataset. At each of these steps, the model updates its parameters.

Results presented in Table 9 show that performance greatly improves, with BLEU scores often above 80 and all significantly above the results without supervision in Table 6, even though the grammars are different here. As a baseline, training only with the supervised examples is not sufficient (for example for **000000↔000000** the BLEU score with only $1,000$ parallel sentences is 31.49, not 96.03).

| Grammars | aligned sentences | bilingual dictionary |
|---|---|---|
| **000000↔000000** | 96.0 | 72.1 |
| **000000↔011101** | 67.4 | 59.8 |
| **000000↔100000** | 81.5 | 42.7 |
| **000000↔111111** | 84.8 | 28.3 |

Table 9: BLEU scores for different language pairs, when augmented with supervised signal. The column **aligned sentences** corresponds to the addition of supervised training on a parallel dataset of 1,000 sentences (out of 100,000). The column **bilingual dictionary** to the addition of the supervised training on all word pairs (passed in as one-word sentences). Both supervisions lead to better scores than without any supervision (see Table 8, where top BLEU score is 23).

### 9.2 Bilingual Dictionary

Previous results suggest that failure arises at the lexical level, so here we add supervised training on all pairs of aligned words, i.e. the complete bilingual dictionary. Results improve compared to previous results: BLEU scores are now between 28 and 72 (see Table 9), when they were previously below 7 (see Table 6).

Overall then, back-translation works drastically more efficiently when complemented with some supervised signal, either at the sentence level (previous subsection) or at the lexical level (this subsection).

## 10 Experiment 6: Towards Semantics via Lexical Fields

One difference between our artificial languages and natural languages concerns semantics. Currently, it appears that our models roughly generate the correct POS sequence and then each word is randomly sampled, with no regard for semantic dependencies. Some methods have been proposed to address this sampling problem (Arora et al., 2016; Hopkins, 2022a).

To stay as close as possible to our previous experiments, we propose here a first-order approximation of the semantic dependencies between different words in a sentence: each word in a 'content' POS (Noun, Adj, TVerb, IVerb, Verb Comp) is associated with one of $N_c$ lexical fields (respecting morphology, so *bird* and *birds* would be associated with the same lexical field), and each sentence in the corpus is made of words from a single lexical field, or context (we use the two terms interchangeably). Words within a given POS and lexical field are sampled from a power law. This is compatible with results from Guerin et al. (2024) showing that beyond the usual Zipf law for word frequencies, co-occurrences of two pairs also follow a power law, and that in practice very few pairs of words actually co-occur in a single sentence (which is compatible with words belonging to different lexical fields). See examples in Table 10. The idea of this experiment is to see whether the model is capable of picking up this simple semantic cue.

To measure success at capturing the lexical field information, we calculated (i) the proportion of sentences containing only words from the same lexical field, and (ii) for accurately POS-translated sentences, the proportion of words that were translated into a word from the right lexical field.

### 10.1 Experiment 6a: Two balanced lexical fields

In a first sub-experiment, we used $N_c = 2$ lexical fields, and sampled the context of the sentences uniformly across the two contexts. Across 4 training seeds, we systematically obtained a proportion of mono-context

| Sentence | no constraint | lexical field |
|---|:---:|:---:|
| burse sub kurches | ✓ | ✓ |
| hostician sub lurchifies | ✓ | ✓ |
| burse sub lurchifies | ✓ | ✗ |
| hostician sub kurches | ✓ | ✗ |

Table 10: Examples of sentences from the vanilla corpus (without lexical fields constraints) and from the 2 lexical fields corpus. In these examples the words *burse* and *kurches* are a noun and a verb from the first lexical field, while *rheleates* and *lurchifies* are a noun and a verb from the second lexical field. Without the lexical field constraint, all pairs of words can co-occur and we can form four sentences, while the lexical field constraint eliminates the mixed sentences. This constraint does not apply to function words, like *sub* here, which are transparent to lexical fields.

.

sentences of 1. This success may come from the DAE only, which plays the role of language modeling, completing sentences within a given context.

For word-by-word translations, the accuracy was around 35% for 2 training seeds, and around 65% for the other 2 training seeds. This corresponds to the fact that in unsupervised learning, there is no signal to map contexts correctly across languages, since they are permutable within each language. This is thus an example of the rotation situation from Figure 2c. This precision yet is not at ceiling: either 0% or 100%. This indicates that the model fails to learn perfectly this partition of our languages.

### 10.2 Experiment 6b: Two unbalanced lexical fields

In a second sub-experiment, we used $N_c = 2$ lexical fields again, and sampled the context of sentences across the two contexts with an unbalance proportion of 30/70. The idea is to give the model the material to distinguish between contexts and overcome the bi-modal behavior noted above.

The BLEU score was slightly increased compared to the same experiment without lexical fields (see Table 8). The proportion of mono-context sentences remains at one. The proportion of words translated in the right context are, over six training sessions: 55%, 65%, 66%, 70%, 73%, 76%.

A naive baseline consisting in choosing the most frequent context would be at 70%. But this is visibly not what the models are doing (the output sentences are from both contexts). Another naive baseline consisting in choosing the context according to its proportion in the corpus would obtain 58%. The models are (almost always) above that level. In addition, unlike in the case of equally frequent lexical fields, the different trained models now tend to all choose the right mapping between lexical fields, therefore taking advantage of their relative frequencies to map them onto one another. This is both a good result, and a risk: if lexical fields are present in different proportions across cultures/languages, this could create a wrong signal. In sum, the models are, to some extent, able to use semantic frequency cues to map lexical fields across languages.

### 10.3 Experiment 6c: Ten unbalanced lexical fields

We replicated the lexical field experiment with $N_c = 10$ lexical fields, and a number of sentences in each lexical field varying according to a power law with parameter $k = 1.1$ as before, thereby providing finer-grained 'semantic' information. The mono-context sentence proportion remains perfect at 1. The BLEU score increases slightly to 11.92 (see comparative results in Table 8). The proportion of words translated in the right context are: 27%, 28%, 38% and 39%, which is above a random baseline at 20%.

### 10.4 Discussion

Overall, these experiments show that more and more fine-grained semantic information provides key signal for translation alignment, even if it is not fully sufficient in this simple form to make unsupervised back-

translation fully work. The implementation of the semantic signal through a rigid lexical field constraint surely is an oversimplification. Whether more realistic versions would lead to a clearer or a noisier effect on translation quality is hard to predict. We tried evaluating the effect of more complex semantics dependencies using the model of Hopkins (2022b).[4] Such more sophisticated processes did not lead to qualitative improvements, and they come with their own simplifications. In future work, the use of more complex and more realistic models of semantic dependencies could help decide whether the effect of the first-order lexical field constraint is confirmed.

## 11  Conclusion

The back-translation objective is not sufficient to align two sets without supervision, in general. This is true even if it is complemented with additional objectives such as filtering or denoising auto-encoding. The method is successful nonetheless with real languages. This success then is presumably due to similarities between natural languages, which the training method picks up on. But it is not clear which similarities help do that. Through controlled experimentation with artificial languages, we investigate the role of lexical, syntactic and semantic properties.

We find that, when they share the exact same lexica, languages with more similar grammars are easier to translate into one another. Hence, grammatical similarity across languages of the world could be a key to the success of back-translation. But when lexica also vary, syntactic similarities are not sufficient to make back-translation align two languages. Lexical alignments are thus hard to learn by back-translation. What language properties make them learnable? We find that neither anchor points (partially shared vocabulary), nor rich, parallel word frequencies are enough to make back-translation work. Thus, manipulating various lexical and syntactic properties only, we find that some supervision signal is critical to support back-translation: through a small set of aligned sentences, or a complete set of aligned words.

Moving to semantics then, we explored how the distribution of word cooccurrences influence the efficiency of back-translation. We used only a crude form of semantics, by implementing lexical fields: different classes of words that never occur within the same sentence (think about: 'clothes', 'sock', 'shoe', 'shirt' and 'astrophysics', 'interstellar', 'electromagnetic'). We find that unsupervised back-translation models are able to pick-up on this (coarse) semantic signal to find a better alignment. We conclude that the success of back-translation is probably due to an even richer semantic parallelism across languages, above and beyond their lexical and syntactic similarities.

In the future, one would like to study more subtle semantic properties then. Currently, the semantic information we implemented is both coarse-grained, and not completely decisive. One would thus like to test whether more realistic semantic information improves the system even more, or makes it collapse again. And subtle properties could be investigated. For instance, selectional restrictions on verbs may play a role in shaping text distributions, above and beyond syntactic constraints, in a way that may help induce alignment across languages in an unsupervised setting. Future work should thus explore realistic semantic distributions, while maintaining the experimental control that artificial languages provide (Hopkins, 2022a). This will help understand what in natural languages makes unsupervised back-translation reach so much success, despite its *a priori* theoretical insufficiency.

### Broader Impact Statement

We hope that our work can have two impacts, navigating between engineering and scientific communities. In one direction, systematic investigations we present can help evaluate and improve applied translation system, and in particular for low resource languages where supervision is not an option. Conversely, we take advantage of the engineering success of back-translation, to unearth what natural languages are made of, how they spontaneously align with one another.

---

[4]The idea is to introduce a hierarchical Pitman-Yor processes to generate semantic dependencies. In a nutshell it makes the generation of the sentences dynamical by modifying the probability of the next constituents based on the ones already generated. For instance after the occurrence of the verb *eat*, the probability to generate the word *water* decreases and that of the word *food* increases.

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
