# OpenReview forum: "The Impact of Syntactic and Semantic Proximity on Machine Translation with Back-Translation"
_TMLR — Accepted by TMLR_

### Review · Reviewer_JW9t · 2024-04-25

**Summary Of Contributions:**

This paper evaluates the performance of back translation approaches for neural machine translation through controlled experiments with artificial languages as an attempt to theoretically uncover what properties of languages allow back translation to work at all as an algorithm. Specifically they construct and look at languages that differ in word frequency distributions and syntactic structure, allowing them to evaluate how the similarity of lexical fields across languages can allow better back translation. They discuss how this can help understand which languages, based on their syntactic and lexical properties might help develop cross-linguistic tools for better translation methods.

**Audience:**

Yes

**Claims And Evidence:**

Yes

**Requested Changes:**

A discussion section about a mapping between the grammars/artificial languages and real languages in the world would be helpful to understand the implications on real translation tasks.

A table that gives an overview of this mapping with example languages would be a great help to the community and future work drawing off of this paper.

Having one more metric that is an alternative to BLEU (especially for the lexically similar languages) would be good to see to verify that the trend of results hold up.

A discussion section about the implications of the metrics used and how they favour some language pairs over others would be helpful.

**Strengths And Weaknesses:**

This paper is clearly written and explained and the hypotheses tested are well formulated and valid tests of the back translation method.

The experimental methodology is sound and carried out in detail for the artificially generated languages the authors test. These languages and experimental results allow us to a deeper understanding of the effect of variation/similarities in languages and how this can affect models that attempt to back translate between languages.

The generation procedure of artificial languages, details of grammar, sizes of datasets produced etc., are all well documented and easy to replicate.

Weakness of evaluation: the main metric to score translations in this paper is BLEU scores of generations, and although this is widely and most commonly used for machine translation evaluations, there are several papers that highlight the deficiencies of this metric (see here: https://aclanthology.org/E06-1032.pdf) and alternatives to it (see here: https://aclanthology.org/2021.triton-1.6.pdf). Especially since some of the experiments hinge on lexical similarity between languages, it would be worth discussing alternative metrics for some of this paper (e.g., why the limitations of this metric might result e.g., in higher results for the lexically similar languages and highlighting potential differences that might occur if the other metrics that are less biased towards simple n-gram overlap were to be used.)

Weakness of lexical/similarity hypotheses: the similarities between real languages are often due to more subtle semantically aligned properties rather than just pure lexical similarity that is evaluated here. Although the simple case of lexical overlap or anchor points is important to evaluate, it is worth also attempting to evaluate the more nuanced similarities that occur in real languages (or atleast posit how this might influence back translation methods).

In order to understand how the results form these artificial languages can be mapped to real languages that we might want to apply these translation methods to, it would be useful to have a table or paragraph that outlines how pairs of real languages fall into the categories of hypotheses framed here (e.g., which pairs of languages have some high percentage of lexical overlap, anchor points, syntactic similarity and so on). Future work that draws on this paper can then attempt to empirically evaluate if real translation datasets/models on this task also have similar effects as is seen by the languages here.

---

> ### Author Response · Authors · 2024-07-24
>
> Thank you very much for this very relevant review!
>
> **Question about BLEU**
>
> It is true that BLEU is flawed in many ways: it cannot handle paraphrases and synonyms, does not take meaning into account directly, it is invariant by n-grams permutations, it has a low correlation with human judgment, etc. The situation is as follows for our specific setup (and we can include such a discussion in a revision):
> 1) Our setup does not include paraphrases nor synonyms, hence these limitations of BLEU do not apply.
> 2) Similarly, meaning is not directly relevant. This could lead to an overestimation of BLEU, but our main arguments rely mainly on low scores, hence these arguments remain.
> 3) Finally we compute BLEU scores for n-grams ranging from 1 to 4. Hence, it helps capture syntactic structure (order constraints) and this mitigates the risk of bigrams permutations insensitivity, as illustrated in https://aclanthology.org/E06-1032.pdf.
>
> We could also consider (and discuss) alternatives to BLEU. Alternatives to BLEU include two families: the automatic language agnostic metrics like BLEU, METEOR or WER, and the LLM-based metrics like BERTScore, MetricX, Comet, Unite, BLEURT, etc. We cannot use the second ones, since we are dealing with artificial languages. WER is known to be worse than BLEU. We could use METEOR, but it remains imperfect as stated in https://aclanthology.org/2021.triton-1.6.pdf and not much better than BLEU. We can try with it, even though we are convinced that it will not change the conclusions as the results are expected to be close to BLEU, and our effects are not small.
> Finally, note that we already complemented BLEU with self-crafted metrics for our particular translation setup. We looked manually at precision & recall of each word translation based on its POS and frequency (see Figure 3) to see that in fact only a handful of words have a non-zero score and we computed the entropy of the model translation per POS (Table 7) to demonstrate that in fact the model does roughly a 1-to-1 mapping, just an erroneous one.
> Put together, we hope that these considerations about our setup and additional measures help deliver a complete picture of the results, despite the limitations of BLEU.
>
> **Mapping between artificial languages and natural languages**
> We agree that adding a Table would help understand the relevance of the grammar and grammar differences, and the impact of our work. There already exist examples in the original paper, and we can include such information here as well. More generally, we can discuss the fact that switches of the sort are a good approximation of some differences across languages.
>
> Again thank you very much for this very relevant review!

---

### Review · Reviewer_5UWs · 2024-06-03

**Summary Of Contributions:**

This paper uses artificial languages based on context-free grammars to inspect the behavior of back-translation for unsupervised neural machine translation. With a shared vocabulary, using different grammars leads to various degrees of success, generally with lower performance for major ordering changes between the two languages. When using identical grammars, but fully disjoint vocabularies, translation performance is very poor even if round-trip translation is often successful, with the model learning the wrong mapping between tokens. Using shared words or non-uniform word frequencies is not sufficient, while adding some supervised signal is helpful. Using lexical fields, i.e. non-overlapping word subsets, hints that richer semantic dependencies may be necessary for unsupervised NMT.

**Audience:**

Yes

**Claims And Evidence:**

Yes

**Requested Changes:**

**Questions**

P1. Why is creating parallel datasets "sometimes an impossible task"?

P5. Why 1374 words?

P5. Why share 3 layers out of 4? In Lample et al. (2018c), it seems that all parameters are shared.

**Comments**

In the related work section, Lample et al. (2018a) is presented as a follow-up to Artetxe et al. (2018), but I believe both papers were contemporaneous.

**Typos**

P2. exerpimental
P13. unbalance

**Strengths And Weaknesses:**

**Strengths**

- The paper presents a series of controlled studies to understand what may enable the success of back-translation for unsupervised NMT.

- The paper challenges assumptions about the success of unsupervised NMT.

**Weaknesses**

- 100,000 sentences seem low for unsupervised NMT. It is not fully clear if and how some of the findings would differ with larger datasets.

- Semantic properties are only explored at a coarse level, and more complex semantic properties may be difficult to simulate with artificial languages.

---

> ### Author Response · Authors · 2024-07-24
>
> Thank you very much for your review!
>
> **Add more sentences to our corpus**
>
> We did try with a 1,000,000 sentences corpus in several scenarios and it did not lead to any significant improvement so to save time we kept running our experiment with only 100,000 sentences.
>
> **Including complex semantic properties**
>
> We did try using more realistic artificial languages like https://aclanthology.org/2022.conll-1.7.pdf which introduces hierarchical Pitman-Yor processes to generate semantic dependencies (in a nutshell: after the occurrence of the verb “eat”, the probability to generate the word “water” decreases and that of the word “food” increases). Such more sophisticated processes did not lead to any improvement. This led us to study in independent work the semantic properties of natural languages (eg the distribution of the cooccurrences), and this is a new work of its own. We do wish to push in that direction, now that it is established that relatively simple artificial languages are not enough to make back-translation work. Which properties do make back-translation works is the question that is now open and addressable from there.
>
> **About the other questions**
>
> *Why is creating parallel datasets "sometimes an impossible task?*
> It is sometimes an impossible task as recent large language models require too much data to train, hence relying only on parallel data would need too much work and hence money, especially for low-resource languages.
>
> *Why 1374 words?*
>
> We used 1374 words to obtain a replication of https://aclanthology.org/2021.acl-long.38/. Even with so few words it did not work, so we did not try with a larger vocabulary and focused on finding linguistic properties that would make it work.
>
> *Why share 3 layers out of 4? In Lample et al. (2018c), it seems that all parameters are shared*
>
> In the Github of Lample et al. https://github.com/facebookresearch/UnsupervisedMT/tree/main default parameters, to reproduce the paper results, only 3 out of 4 encoder layers were shared so we did the same.
>
> Thank you very much for pointing out typos and for the remarks about the related work section, we will indeed include your remarks.
>
> Again thank you very much!

---

### Review · Reviewer_juUp · 2024-07-16

**Summary Of Contributions:**

This paper aims to better understand **why** on-the-fly back-translation works well in the case of unsupervised machine translation, even when there are no theoretical guarantees for their success. To that end, the paper used an artificial language setup where it is possible to disentangle the impact of several different factors on back-translation's success in the context of unsupervised machine translation: (i) the extent to which the word frequency distributions are similar across the two languages; (ii) the extent to which the vocabularies are shared / there are lexical overlaps; (iii) the extent to which the syntax between the two languages differs; and (iv) the extent to which crude semantic signals (in particular, lexical fields) are similar across the two languages.

The paper began by demonstrating that back-translation is not theoretically guaranteed to succeed in the case of unsupervised machine translation. Furthermore, while prior work has hypothesized the importance of similar syntactic structures, word frequency, and anchor points (i.e. the fact that numbers tend to have similar surface strings across different languages, and can therefore serve as anchor points for other words different languages), this paper finds evidence to the contrary. While the paper finds that shared syntactic structures improve back-translation performance, syntactic similarity alone is insufficient to make back-translation work well in the presence of strong lexical differences. Furthermore, neither anchor words nor similar word frequencies are sufficient to make back-translation succeed, although some degree of shared semantic parallelism (here in the form of **lexical fields**) is helpful.

The paper concludes that back-translation success can likely be attributed to the richer semantic parallelism across languages. Furthermore, *some* degree of supervision signal is critical for back-translation success for unsupervised machine translation, whether through a small number of aligned sentences (i.e. a small parallel corpus) or a complete set of aligned words (i.e. a good bilingual dictionary, whether provided from an existing dictionary or learned through bilingual lexical induction).

**Audience:**

Yes

**Broader Impact Concerns:**

No broader impact concerns from my side, and I am happy with the broader impact statement as it is written on the paper.

**Claims And Evidence:**

Yes

**Requested Changes:**

1. **Recommended**: A discussion around what the paper's findings mean for designing better unsupervised machine translation system through back-translation (e.g. that a good dictionary / some small amount of supervision is important, etc.).

2. **Recommended**: More discussion and space dedicated towards explaining the lexical field experiment, ideally with examples, given that the paper highlights the importance of semantic similarity across languages.

3. **Recommended**: A discussion around how realistic the lexical field assumption is in real languages.

4. **Recommended**: A discussion around whether or not pre-training and cross-lingual word embeddings would likely mitigate the issues highlighted here (e.g. helping to find alignment between different words in different languages, etc.).

**Strengths And Weaknesses:**

# Strengths

1. The paper addresses an important question of **why exactly** back-translations work in the case of unsupervised machine translation, even though there are no theoretical guarantees for their success. This line of work is important for the community to better understand the factors that drive back-translation success, and understand for what kinds of language pairs back-translation would likely succeed (or not succeed, e.g. when there is a substantial amount of lexical differences).

2. The paper is rigorous and extensive in terms of its analysis and methodology, which is done through synthetic language pairs, where one can control for the different factors (e.g. degree of syntactic & lexical overlap, etc.). I appreciate that the paper covers multiple different factors that can affect back-translation performance, such as syntactic & lexical & semantic overlap, etc.

3. The paper is very clear and well-written, and includes a comprehensive overview of both unsupervised machine translation and back-translation. This would be useful for readers that have not worked directly on this line of work.

4. The findings would likely be of interest to the broader community.

# Weaknesses

1. While the findings are interesting, it is not immediately clear how to apply the findings to make back-translation work better for unsupervised machine translation. I do think that the paper can make some recommendations based on its findings (e.g. that some degree of supervision is necessary, so having high-quality bilingual dictionary or a small amount of parallel corpus would be very helpful, or that having cross-lingual word embedding that puts similar words in different languages in the same vector space is essential etc.). It would be nice to see the implications of the paper's findings spelled out more clearly, so that the broader community can build on the findings to design better-performing approaches.

2. Not enough space is dedicated to the semantic similarity section through lexical fields (Section 10, Experiment 6, page 12). This is an important part of the paper, because this shared semantic information helps translation alignment to a good extent (more so than other factors like syntactic similarity, for example). A more extensive description of lexical fields and what is exactly being done, ideally along with some examples, would be very helpful for the reader.

3. I am not entirely certain about how realistic the lexical field assumptions are in real languages (in particular, the assumption that each sentence in the corpus is made of words from a single lexical field, bottom of page 12). Some discussion around this would be much appreciated.

4. A discussion around how pre-training and cross-lingual embeddings would affect the findings would also be very helpful. Are the problems with lexical overlap, etc. can be solved by multi-lingual pre-training on massively multi-lingual data? Or would these likely to persist?

---

> ### Author Response · Authors · 2024-07-24
>
> Thank you very much for your very detailed and positive review. You are right and we should definitely add discussion where you pointed out. Here a some thoughts on your remarks:
>
> 1) We should definitely emphasize the implications of our work on MT. Even if the investigation is not complete because we should explore more semantics properties our results already point out - even if it is not new - that for instance adding a very small amount of supervised signal rather than putting effort on collecting larger amounts of aligned data can largely improve on BT results. And this is even more true for a pair of very different languages. We note however that unsupervised translation does work to a large extent in real conditions, and the puzzle that we raise is that it is hard to understand why (not necessarily that it should be improved upon).
> 2) We agree that Section 10 on lexical fields can be expanded. We will expand it and propose to include more examples, and a Figure to illustrate the method.
> 3) You are right that the assumption of a unique lexical field per sentence is an oversimplification. We see this unrealistic setup as one that could make the task easier than a realistic one. Given that the model fails nonetheless suggests that more realistic lexical fields won’t succeed either. (In future work, as suggested by Reviewer 2 and discussed in the response, we plan to use more complex and realistic semantics dependencies.)
> 4) Pre-training should improve results (or at least fasten) training since at the very least it forces the decoders to generate output that belongs to the correct language, therefore preventing the scenario of Figure 2.b. Here, the training is already done alongside a LM objective (Denoising Auto-Encoding) and this does not suffice to “cure” BT. And similarly, we already use cross-lingual word embeddings as the tokenization and the initialization of the embeddings is done jointly, yet it is not sufficient.
>
> Thank you again for your very helpful comments. We will discuss in the next iteration of the manuscript.

---

> > ### Comment · Reviewer_juUp · 2024-08-15
> > **Thank You for the Response**
> >
> > Thank you for the authors' response. Having read the response to my reviews (which address my concerns) and also the other reviews and the authors' responses to them, I hope that many of the suggestions will be integrated in subsequent versions.
> >
> > I maintain my initial assessment regarding the strengths of the submission, the scientifically interesting findings, and also its relevance for the community.

---

### Decision · Action_Editor_HgWh · 2024-08-26

**Recommendation:** Accept as is

**Comment:**

All the reviewers and myself concur this an interesting and well-executed paper illuminating an important empirical technique that is seldom studied in a controlled setting. The paper is scientifically rigorous, well-written and clear. Reviewers pointed out some of the limitations of the paper that should be addressed in the camera-ready, notably 1) strive to make the paper's findings spelled out more clearly, so that new methods could more easily be designed; and 2) the decision of using the lexical field assumption, which is quite restrictive for real languages. I believe the authors already promised to elaborate and even provide more experiments on 2) for the camera-ready. Overall, this is great work and deserves acceptance.

**Audience:**

Linguistics/natural language processing community.

**Claims And Evidence:**

The paper studies why back-translation works in the case of unsupervised machine translation. Languages with similar grammars are easier to translate into one another when they share the same lexica, highlighting the importance of grammatical similarity in back-translation. If lexica differ however, syntactic similarities alone are insufficient for effective back-translation and a small set of aligned sentences or words is crucial. Then the paper shifts to studying whether semantics is responsible for back-translation success. The study finds that some degree of shared semantic parallelism (here in the form of lexical fields) is helpful. The study concludes that "the success of back-translation is probably due to an even richer semantic parallelism across languages, above and beyond their lexical and syntactic similarities".